# Association between depressive symptoms and cognitive–behavioural therapy receipt within a psychosis sample: a cross-sectional study

Ava Mason ,[1,2] Jessica Irving ,[2] Megan Pritchard,[2,3] Jyoti Sanyal,[3] Craig Colling,[2,3] David Chandran,[2] Robert Stewart[2,3]

¹Division of Psychiatry, University College London, London, UK
²Institute of Psychiatry, Psychology and Neuroscience, King's College London, London, UK
³South London and Maudsley Mental Health NHS Trust, London, UK

**Correspondence to**
Ava Mason;
ava.mason.20@ucl.ac.uk

## ABSTRACT

**Objectives** To examine whether depressive symptoms predict receipt of cognitive–behavioural therapy for psychosis (CBTp) in individuals with psychosis.

**Design** Retrospective cross-sectional analysis of electronic health records (EHRs) of a clinical cohort.

**Setting** A secondary National Health Service mental healthcare service serving four boroughs of south London, UK.

**Participants** 20 078 patients diagnosed with an International Classification of Diseases, version 10 (ICD-10) code between F20 and 29 extracted from an EHR database.

**Primary and secondary outcome measures** Primary: Whether recorded depressive symptoms predicted CBTp session receipt, defined as at least one session of CBTp identified from structured EHR fields supplemented by a natural language processing algorithm. Secondary: Whether age, gender, ethnicity, symptom profiles (positive, negative, manic and disorganisation symptoms), a comorbid diagnosis of depression, anxiety or bipolar disorder, general CBT receipt prior to the primary psychosis diagnosis date or type of psychosis diagnosis predicted CBTp receipt.

**Results** Of patients with a psychotic disorder, only 8.2% received CBTp. Individuals with at least one depressive symptom recorded, depression symptom severity and 12 out of 15 of the individual depressive symptoms independently predicted CBTp receipt. Female gender, White ethnicity and presence of a comorbid affective disorder or primary schizoaffective diagnosis were independently positively associated with CBTp receipt within the whole sample and the top 25% of mentioned depressive symptoms.

**Conclusions** Individuals with a psychotic disorder who had recorded depressive symptoms were significantly more likely to receive CBTp sessions, aligning with CBTp guidelines of managing depressive symptoms related to a psychotic experience. However, overall receipt of CBTp is low and more common in certain demographic groups, and needs to be increased.

## STRENGTHS AND LIMITATIONS OF THIS STUDY

⇒ To our knowledge, this is the first electronic health record (EHR) study to measure how clinical symptomatology predicts cognitive–behavioural therapy for psychosis (CBTp) receipt, providing insight on a large sample into whether individuals who may be more in need of CBTp are more likely to have a session.

⇒ We replicate previous findings of inequalities in gender and ethnicity in real-world CBTp treatment receipt in a large heterogeneous sample.

⇒ The natural language processing approach allows automated processing of EHR text at scale and can evaluate larger samples than manually conducted case note audits; this could therefore be used more routinely to monitor CBTp receipt.

⇒ This study was limited to a single service provider; however, the results identified themes consistent with previous CBTp provision research in other services.

⇒ Analysing EHRs in this way can identify CBTp receipt but is less suited to investigate whether CBTp is offered or not, or to quantify the quality or focus of the sessions. Furthering this, it cannot be used to examine CBTp completion rates and effectiveness.

## INTRODUCTION

There are a variety of cognitive and emotional processes involved in the development of psychotic symptoms,[1] with intense distress emerging early on in the course of the disorder. Content of positive symptoms often mirrors the content of depressive thinking processes,[2] suggesting therapeutic need for individuals experiencing additional depressive symptoms. Specific depressive symptoms that often accompany psychotic disorders are hopelessness, social avoidance and problems in forming relationships.[3] Around 50% of patients with psychosis report having experienced suicidal ideation at least once,[4] and around 40% of individuals with schizophrenia report clinical levels of depression and low self-esteem.[5] Importantly, individuals report these emotional difficulties and resulting social exclusion to be more debilitating than

their psychotic symptoms.[6] Consequentially, individuals' negative appraisal of their psychotic experiences may lead to loss of social goals and increased shame, predicting later hopelessness and postpsychotic depression.[7] This comorbid depression increases the likelihood of having a lower quality of life, function, motivation, poorer social relationships, lower medication adherence and psychotic relapse.[8 9] Therefore, treatment should focus on the psychotic symptoms and the broader distress they produce, building self-esteem, confidence and a sense of self control and purpose.[10] Additionally, focusing on mood symptoms such as self-esteem and pessimism can help differentiate depressive symptoms from negative psychotic symptoms, that often show significant clinical overlap.[5]

It is increasingly recognised that medication alone is inadequate for tackling psychosis symptoms.[11] In the UK, the National Institute for Health and Clinical Excellence[12] has recommended that cognitive–behavioural therapy for psychosis (CBTp) be offered universally to individuals with psychosis. Based on the stress-vulnerability model,[13] CBTp focuses on distress reduction related to hallucinations and delusions, through targeting negative beliefs and improving self-esteem.[14] Sessions often focus on goal setting and emotional issues such as rebuilding one's self, positivity and acceptance.[11] While studies examining characteristics of CBTp show strong evidence that CBTp improves depressive symptoms in the context of psychosis, specifically with long-term reductions in suicidal behaviour,[14 15] service provision of this intervention still falls far short of the universal access recommended.[11]

Considering the impact of targeting these symptoms in CBTp sessions, it is important to monitor receipt of CBTp within psychosis samples. While CBTp provision shows moderate yearly increases (12.8% in 2013 to 14.8% in 2014), the treatment is still only available to a small proportion of individuals,[11] short of NICE universal access recommendations.[12] Previous studies investigating CBTp receipt have conducted time-consuming audits on limited sample sizes; these can be affected by under-reporting. On the other hand, the UK's National Mental Health Minimum Data Set report does not require CBT interventions to be recorded in a given individual's record. Natural language processing techniques (NLP)[16] offer the opportunity to extract this information from free text in electronic health records (EHRs) across large numbers of patients with psychosis, and a recent study developed and applied NLP in this respect, finding higher levels of receipt than reported in previous audit, supported by the high positive predictive value (PPV) and sensitivity of the technique (95% and 96%, respectively).[11]

While studies have examined general CBTp receipt within patients with psychosis, no study has examined a link between depressive symptoms and CBTp receipt.[11] Therefore, we investigated whether depressive symptoms predict CBTp receipt in people with psychosis by applying these previously data extraction techniques to secondary mental healthcare EHRs for a large South London catchment population. Secondary predictors of receipt were type of psychosis diagnosis (schizophrenia, schizoaffective disorder or other schizophrenia spectrum disorder), symptom profiles (negative, manic or disorganisation), general CBT receipt prior to psychosis diagnosis, comorbid depression, anxiety or bipolar diagnosis and socio-demographic factors (ethnicity, gender and age).

## METHODS

For this study, we extracted data on individuals with a diagnosis of a recognised schizophrenia spectrum diagnosis from the case registry of the South London and Maudsley National Health Service Foundation Trust (SLaM). This is a large secondary care mental healthcare provider, serving around 1.3 million residents in Croydon, Lambeth, Lewisham and Southwark. SLaM care covers all specialist mental healthcare, including early intervention services, liason and crisis teams and community and inpatient services. EHRs have been used for all SLaM services since 2006, with the Clinical Record Interactive Search system (CRIS) being established in 2008 to facilitate the retrieval of deidentified data from these records of patients previously or currently receiving mental healthcare from SLaM.[17] The source EHR contains unstructured free text fields from correspondence, personal histories, mental health examinations and management plans, as well as structured fields for coding demographic information, like age and ethnicity. Implementing data from all these fields reduces selection bias of using only specific sources of information from the EHR. Consequently, a large programme of work has developed a range of NLP algorithms over the last decade, whose detailed descriptions and performance data are contained in an open-access catalogue.[18]

We extracted data for all individuals receiving SLaM care between January 2007 and June 2020 with a primary diagnosis of an International Classification of Diseases, version 10 (ICD-10)-defined schizophrenia spectrum disorder (F20–F29) and above the age of 18 at the time their original referral was accepted. The index date for covariate definitions was the date of the first diagnosis within this grouping. Individuals may have been active within the service before their index date, allowing us to extract data on prior CBT receipt. The sample was restricted to those with data on all variables.

Ethnicity, age at referral and gender were also extracted. Ethnicity was categorised into six groups for analysis: 'white British' (British), 'white other' (Irish or any other white background), 'black' (Caribbean, African or any other black background), 'Asian' (Indian, Bangladeshi, Pakistani, Chinese or any there Asian background), 'other/mixed' (white and Asian, white and black Caribbean, white and black African, any other ethnic group) and 'not stated'.

Diagnosis was categorised into three subgroups of schizophrenia (ICD-10 codes F20.0–F20.9), schizoaffective disorder (F25.0–F25.9) and other schizophrenia

spectrum disorder (F21, F22.0–F22.9, F23.0–F23.9, F24, F28 and F29). Within the data collection period, secondary diagnosis of depression (ICD-10: F32 or F33), anxiety (ICD-10: F40 or F41), or bipolar disorder (ICD-10: F31) were also extracted from structured field data.

NLP algorithms for each specific symptom were used to identify recorded symptom profiles within participants. Symptoms were categorised as depressive, positive, negative, manic or disorganisation. These symptoms had been categorised a priori by developers of the original independent symptom NLP algorithms. As symptoms could be labelled in more than one category during analysis, multicollinearity tests using the R function vif() within the (car package) were undertaken to avoid issues with overlapping predictor variables. All variables were included due to their VIF values being below five. However, positive symptoms were excluded from regression models using categorical symptom variables (having at least one mention within HER), as this factor variable only had one level, due to all participants having at least one positive symptom. The overall symptom list and subsequent recoding can be found in table 1. Presence of at least one mention of any symptom in the five categories was computed as a binary variable (0/1).

The date of the first and last general CBT session before the index date was extracted. This was coded as a binary variable, with individuals in the 'Prior CBT' receipt group having at least one session date mention prior to their index date. This was included as a predictor to adjust for previous experience of the specific CBT intervention. Mentions were extracted using the same NLP tool as the CBTp outcome measure mentioned subsequently.

The primary outcome was CBTp receipt, identified using a combination of structured fields and NLP.[16] The NLP algorithms for general CBT has high PPV and sensitivity,[12] consistent with other NLP algorithms such as medication dose and diagnosis.[19] The date of the first CBTp session on or after the index date was extracted and computed as a binary variable, so that individuals in the 'CBTp receipt' group had at least one CBTp session mention after the index date.

## Statistical analysis

To avoid overfitting, we followed the 'one in ten' rule, whereby one predictor can be measured for every 10 events. As the data included 1647 CBTp events, our study was able to include all 12 predictors within the same regression model.

All statistical analyses were conducted using R (V.1.3.9). Descriptive statistics for demographic and clinical variables are reported as frequencies for categorical variables and means and SD for the continuous variable (age at referral). $\chi^2$ tests were also calculated for categorical variables, and t-test for age to measure between-group differences in those with/without CBT receipt. Descriptive statistics were also provided for yearly CBT prior to index date and CBTp receipt post index date within the data extraction time period (2007–2020).

**Table 1** Classification of symptom predictors

| Symptom | Symptom label |
| --- | --- |
| Aggression | Positive |
| Agitation | Positive |
| Anergia | Depressive/negative |
| Anhedonia | Depressive/negative |
| Apathy | Depressive/negative |
| Arousal | Manic |
| Blunted affect | Depressive/negative |
| Circumstantiality | Disorganisation |
| Delusions | Positive |
| Derailment | Disorganisation |
| Disturbed sleep | Depressive/manic |
| Elation | Manic |
| Emotional Withdrawal | Negative |
| Flight of ideas | Disorganisation |
| Formal thought disorder | Disorganisation |
| Grandiosity | Manic |
| Guilt | Depressive |
| Hallucinations (auditory) | Positive |
| Helplessness | Depressive |
| Hopelessness | Depressive |
| Hostility | Positive |
| Insomnia | Depressive/manic |
| Irritability | Manic |
| Paranoia | Positive |
| Persecutory ideation | Positive |
| Poor appetite | Depressive |
| Poor concentration | Depressive |
| Poor motivation | Depressive |
| Poverty of speech | Negative |
| Poverty of thought | Negative |
| Social withdrawal | Negative |
| Suicidal ideation | Depressive |
| Tangentiality | Disorganisation |
| Tearfulness | Depressive |
| Thought block | Disorganisation |
| Worthlessness | Depressive |

Binary logistic regression was used to examine the association between depressive symptoms and receipt of at least one CBT session in the whole sample. For this, three regression models were analysed. Model 1 was an unadjusted model with only depressive symptoms as the predictor variable. Due to significant provision differences seen in previous CBTp studies,[11] model 2 (partially adjusted model), adjusted for sociodemographic variables (age at referral, ethnicity, gender), primary diagnosis group and presence of a comorbid diagnosis (anxiety,

depression and bipolar disorder). Model 3 (fully adjusted model) also adjusted for prior CBT receipt before the index date (first psychosis diagnosis date) and symptoms mention (manic, negative and disorganisation symptoms). Positive psychotic symptoms were not included in these models, as individuals all had at least one mention within their case notes.

As the primary aim of the study was to investigate depressive symptoms as a predictor of CBTp receipt, we also split the depressive symptoms category into the 15 specific depressive symptoms applications within the whole sample. Model 4 was an unadjusted model with the 15 symptoms as predictor variables. Model 5 was a fully adjusted model that adjusted for all the variables in model 3. We also conducted a sensitivity analysis to investigate how results were affected by overlap of negative or depressive symptom annotations, by removing negative symptoms as a predictor from the logistic regression model.

Additionally to measuring whether individual depressive symptoms could predict CBTp receipt, we also also measured whether overall depression severity predicted CBTp receipt. These logistic regression models involved converting depressive, disorganised, manic, positive and negative symptoms into a continuous variable, whereby severity reflected the number of different individual symptoms mentioned within each symptom construct. This allowed for positive symptoms to also be included within regression models. Model 6 was an unadjusted model, with depressive symptom severity as a predictor of CBTp receipt. Model 7 and model 8 were partially and fully adjusted models, controlling for the same variables as models 2 and 3, except categorising symptoms as the continuous rather than categorical variable.

Lastly, to compare differences in the general sample with those with the top 25% quantity for depressive symptoms, we conducted two further regression models. This subsample analysis was conducted to examine predictors of CBTp receipt where a clear clinical indication was present, supplementing the overall findings. Model 9 partially adjusted for sociodemographic factors, diagnostic group and comorbid diagnosis and model 10 fully adjusted for prior CBT, negative and disorganisation symptoms additionally. This group all had at least one manic and psychotic symptom, so these variables were not included in the model.

### Patient and public involvement
The Clinical Record Interactive system as a data resource was developed and is run with extensive patient involvement. However, this particular analysis did not involve patients in its design or implementation.

## RESULTS
### Participants
The cohort comprised 20 078 individuals with the inclusion diagnoses, 1647 (8.2%) of whom received at least one

session of CBTp after their first diagnosis date. The mean age of the cohort was 42.4 years (SD=16.5). Distribution frequencies for all categorical variables can be found in table 2. $\chi^2$ test results represented in this table compared those with or without CBTp receipt. All mentioned variables showed significant between-group differences at p<0.001 apart from gender (No CBTp group females=41.4%, CBTp delivery group females=43.5%; $\chi^2$=2.75, p=0.097). These significant variables include depression diagnosis ($\chi^2$=87.36), bipolar diagnosis ($\chi^2$=71.94), anxiety diagnosis ($\chi^2$=118.28) and prior CBT receipt ($\chi^2$=497). Additionally, the Welch two sample t-test found significant between-group differences in age (t=15.34, p<0.01). Where those who had received CBTp had a lower mean age (M=33.12 SD=11.5) compared with those who did not (M=35.88, SD=13.08). The significant results confirmed the need for further analysis through the regression models. Positive psychotic symptoms were excluded from $\chi^2$ and regression analysis, as all patients had at least one positive psychotic symptom.

### CBT receipt
The descriptive results shown in table 3 and online supplemental figure 1, suggest that there is a low prevalence of both prior CBT and CBTp postdiagnosis across the years, with receipt reducing in recent years (2019–2020) compared with earlier years (2007) of the data extraction period.

### General depressive symptom mention regression analysis
Results from the unadjusted (model 1), partially adjusted (model 2) and fully adjusted regression (model 3) are displayed in table 4. Regression model 1 found that general mention of at least one of 15 potential depressive symptoms significantly predicted CBTp receipt. Regarding models 2 and 3, individuals with at least one depressive, negative or disorganisation symptom mention, being of female gender, white ethnicity, prior CBT receipt and presence of a comorbid affective disorder independently positively associated with CBTp receipt.

### Regression analysis with individual depressive symptoms
Results from the unadjusted (model 4) and fully adjusted (model 5) regression analyses for each of the 15 individual depressive symptoms are displayed in table 5 (N=20 078). Each symptom refers to presence of at least one mention in the patients notes compared with no mention. While all variables were significant in the unadjusted model at p<0.001, the fully adjusted model reduced the significance of suicide ideation (p<0.01) and disturbed sleep (p<0.01), with anhedonia, anergia, apathy and blunted affect becoming non-significant (p>0.05).

### Sensitivity analysis
The non-significant results of certain depressive symptoms (anhedonia and anergia) may have been due to their inclusion within the negative symptom category, causing over-adjustment of the model. To test this, sensitivity analysis was conducted, where the fully adjusted

**Table 2** Distribution frequencies on baseline demographics and diagnoses split by CBTp receipt and primary diagnosis group

| | No CBTp delivery (n=18 431) | CBTp delivery (n=1647) | $\chi^2$ tests |
|---|---|---|---|
| Ethnicity | | | $\chi^2$=100.57* |
| White British | 30% (5516/18431) | 32.8% (540/1647) | |
| White other | 10.4% (1908/18431) | 8.5% (140/1647) | |
| Black | 36.5% (6719/18431) | 41.7% (687/1647) | |
| Asian | 6.5% (1193/18431) | 5.2% (86/1647) | |
| Other/mixed | 9.8% (1808/18431) | 10.5% (173/1647) | |
| Not stated | 7.0% (1287/18431) | 1.3% (21/1647) | |
| Gender | | | $\chi^2$=2.75 |
| Female | 41.4% (7636/18431) | 43.5% (717/1647) | |
| Male | 58.6% (10795/18431) | 56.5% (930/1647) | |
| Bipolar diagnosis | 4.4% (810/18431) | 9.0% (149/1647) | $\chi^2$=71.94* |
| No diagnosis | 95.6% (17621/18431) | 91.0% (1498/1647) | |
| Depression diagnosis | 7.4% (1373/18431) | 14.0% (230/1647) | $\chi^2$=87.36* |
| No diagnosis | 92.6% (17058/18431) | 86.0% (1417/1647) | |
| Anxiety diagnosis | 2.4% (441/18431) | 7.0% (115/1647) | $\chi^2$=118.28* |
| No diagnosis | 97.6% (17990/18431) | 93.0% (1532/1647) | |
| Prior CBT | 3.1% (573/18431) | 14.4% (237/1647) | $\chi^2$=497* |
| No prior CBT | 96.9% (17858/18431) | 85.6% (1410/1647) | |

*P<0.001.
CBT, cognitive–behavioural therapy; CBTp, CBT for psychosis.

regression (model 3) did not include negative symptoms as a covariate. While all significant variables remained significant, non-significant results for anhedonia and apathy were still found. Therefore, we report the fully adjusted model with negative symptoms as a variable for both grouped and individual depressive symptom associations.

**Table 3** Distribution frequencies on CBT receipt (prior to diagnosis) and CBTp receipt (postdiagnosis) per year of data extraction

| Year | CBT prior | CBT post | All CBT |
|---|---|---|---|
| 2007 | 130 | 81 | 211 |
| 2008 | 89 | 146 | 235 |
| 2009 | 59 | 111 | 170 |
| 2010 | 48 | 107 | 155 |
| 2011 | 37 | 105 | 142 |
| 2012 | 39 | 96 | 135 |
| 2013 | 32 | 128 | 160 |
| 2014 | 25 | 143 | 168 |
| 2015 | 24 | 150 | 174 |
| 2016 | 29 | 115 | 144 |
| 2017 | 16 | 127 | 143 |
| 2018 | 16 | 114 | 130 |
| 2019 | 15 | 153 | 168 |
| 2020 | 2 | 71 | 73 |
| Total | 561 | 1647 | 2208 |

CBT, cognitive–behavioural therapy; CBTp, CBT for psychosis.

### General depressive symptom severity regression analysis

Results from the unadjusted (model 6), partially adjusted (model 7) and fully adjusted regression (model 8) are displayed in table 6. Regression model 6 found that depression symptom severity significantly predicted CBTp receipt. Regarding models 7 and 8, depression symptom severity, positive symptom severity, anxiety diagnosis, and being of older age or being of white ethnicity independently positive predicted CBTp receipt. Within model 7, being female also positively increased likelihood of CBTp receipt. Within model 8, negative symptom severity and prior CBT significantly predicted CBTp receipt additionally.

### Depressive symptom regression analysis within the top 25% number of depressive symptoms

This sample comprised individuals with the top 25% number of depressive symptoms (5018 patients), defined to reflect those who might reasonably expect to receive CBT. The sample characteristics and regression analysis can be seen in table 7. Results from the partially adjusted (model 9) and fully adjusted regression (model 10) are displayed in table 7. Table 4 finds that general mention of at least one of 15 potential

**Table 4** Unadjusted, partially and fully adjusted logistic regression models for CBTp receipt (regression models 1, 2 and 3) with categorical symptom measures

| | N (%) | Unadjusted OR (95% CI) | Partially adjusted OR (95% CI) | Fully adjusted OR (95% CI) |
|---|---|---|---|---|
| Depressive symptoms | | | | |
| 1+depressive symptom mention | 18 286 (91.1) | 3.78 (2.94 to 4.96)*** | 3.42 (2.58 to 4.60)*** | 2.00 (1.10 to 3.20)*** |
| Bipolar diagnosis | | | | |
| Has f31 diagnosis | 959 (4.80) | | 0.52 (0.33 to 0.71)*** | 0.32 (0.12 to 0.52)*** |
| Depression diagnosis | | | | |
| Has f32 diagnosis | 603(80) | | 0.52 (0.36 to 0.67)*** | 0.33 (0.16 to 0.49)** |
| Anxiety diagnosis | | | | |
| Has f40/41 diagnosis | 556 (2.80) | | 0.89 (0.66 to 1.11)*** | 0.73 (0.49 to 0.97)*** |
| Age | N/A | | −0.03 (-0.04 to -0.03)*** | −0.03 (-0.03 to -0.02)*** |
| Gender | | | | |
| Male | | | Reference category | |
| Female | 8353 (41.60) | | 0.20 (0.09 to 0.31)*** | 0.20 (0.10 to 0.32)*** |
| Ethnic group | | | | |
| White British | 6056 (30.10) | | Reference category | |
| White other | 2048 (10.20) | | −0.40 (−0.60 to -0.21)*** | −0.37 (−0.57 to -0.17)*** |
| Black | 7406 (36.90) | | −0.16 (−0.28 to −0.04)** | −0.24 (−0.36 to -0.11)*** |
| Asian | 1279 (6.40) | | −0.49 (−0.74 to −0.26)*** | −0.50 (−0.75 to −0.27)*** |
| Other/mixed | 1981 (9.90) | | −0.21 (−0.40 to −0.02)** | −0.18 (−0.37 to −0.01)* |
| Not stated | 1308 (6.50) | | −1.75 (2.23 to −1.22)*** | −1.52 (−2.00 to −1.10)*** |
| Primary diagnosis | | | | |
| Schizophrenia | 9845 (49.00) | | Reference category | |
| Schizoaffective disorder | 2142 (10.70) | | 0.04 (−0.13 to 0.21) | 0.01 (−0.17 to 0.19) |
| Other schizophrenia spectrum | 8091 (40.30) | | −0.10 (−0.22 to 0.01)* | −0.02 (−0.14 to 0.10) |
| Negative symptoms | | | | |
| 1+ negative symptom mention | 13 169 (65.60) | | | 0.75 (0.59 to 0.92***) |
| Manic symptoms | | | | |
| 1+ manic symptom mention | 17 945 (89.40) | | | 1.24 (0.70 to 1.87)*** |
| Disorganisation symptoms | | | | |
| 1+ disorganisation symptom mention | 11 513 (57.30) | | | 0.31 (0.18 to 0.44)*** |
| CBT prior | | | | |
| 1+ prior CBT session | | | | 1.29 (1.12 to 1.46)*** |

Unadjusted (model 1): depressive symptom as a predictor with no adjusted covariates.
Partially adjusted (model 2): results were adjusted for age, ethnicity, gender, diagnostic group, f31 diagnosis, f32 diagnosis, f40/41 diagnosis.
Fully adjusted (model 3): results were adjusted for age, ethnicity, gender, diagnostic group, f31 diagnosis, f32 diagnosis, f40/41 diagnosis, depressive symptoms, prior CBT, negative symptoms, disorganisation symptoms, manic symptoms.
*P<0.05, **p<0.01, ***p<0.001.
CBT, cognitive–behavioural therapy; CBTp, CBT for psychosis; NA, not available.

depressive symptoms significantly predicted CBTp receipt. Regarding model 9, we found that individuals with at least one depressive, negative or disorganisation symptom mention, being of female gender, white ethnicity, prior CBT receipt and presence of comorbid bipolar disorder were positively associated with CBTp receipt.

## DISCUSSION

We believe that this is the first study to examine the relationship between clinical symptomatology and CBTp receipt within a sample of people with psychosis in a naturalistic community setting. In general, only 8.2% of individuals received CBTp within the 13-year time frame of the study, showing the low prevalence of receipt

**Table 5** Unadjusted and fully adjusted logistic regression models for CBTp receipt with individual depressive symptoms as covariates (regression models 4 and 5) for the overall sample

| | N (%) | Unadjusted OR (95% CI) | Fully adjusted OR (95% CI) |
|---|---|---|---|
| Hopelessness | 7345 (36.60) | 4.81 (4.3 to 5.40)** | 1.45 (1.26 to 1.66) |
| Helplessness | 3124 (15.60) | 4.03 (3.62 to 4.50)** | 1.55 (1.37 to 1.76)** |
| Suicide ideation | 9451 (47.10) | 4.11 (3.66 to 4.63)** | 1.25 (1.09 to 1.44)* |
| Poor appetite | 8044 (40.10) | 3.31 (2.97 to 3.68)** | 1.28 (1.13 to 1.45)** |
| Poor motivation | 8630 (43.00) | 4.34 (3.87 to 4.86)** | 1.43 (1.24 to 1.64)** |
| Insomnia | 6870 (34.20) | 3.74 (3.35 to 4.15)** | 1.4 (1.24 to 1.58)** |
| Disturbed sleep | 16 667 (83.00) | 15.3 (10.16 to 22.8)** | 2.76 (1.5 to 5.08)* |
| Poor concentration | 12 289 (61.20) | 8.16 (6.81 to 9.77)** | 2.33 (1.9 to 2.85)** |
| Anhedonia | 4047 (20.20) | 2.9 (2.61 to 3.22)** | 0.97 (0.85 to 1.10) |
| Anergia | 873 (43.50) | 2.63 (2.20 to 3.15)** | 0.98 (0.80 to 1.20) |
| Apathy | 4149 (20.70) | 2.21 (1.98 to 2.46)** | 0.93 (0.82 to 1.05) |
| Guilt | 8178 (40.70) | 4.6 (4.1 to 5.15)** | 1.49 (1.30 to 1.70)** |
| Tearfulness | 10 951 (54.50) | 3.87 (3.41 to 4.39)** | 1.22 (1.05 to 1.42)* |
| Blunted affect | 6889 (34.30) | 2.66 (2.41 to 2.95)** | 0.91 (0.80 to 1.03) |
| Worthlessness | 2921 (14.50) | 3.94 (3.53 to 4.40)** | 1.37 (1.21 to 1.56)** |

*P<0.01, **p<0.001.
CBTp, cognitive–behavioural therapy for psychosis.

despite current clinical guidelines. This finding shows a reduction in CBTp provision compared with previous studies in 2013 (12.8%) and 2014 (14.8%),[11] which was further supported by the descriptive frequency results, showing a drop in both CBT and CBTp receipt in recent years. This requires further examination, as it is unclear why receipt is decreasing considering the importance of CBTp mentioned within NICE universal access recommendations.[12]

Ninety-one per cent of patients had at least one recorded depressive symptom mention. Individuals with at least one depressive symptom mention were two times more likely to have at least one CBTp session in the fully adjusted model (table 4), suggesting that the minority who don't present with any depressive symptoms are very unlikely to receive CBTp. This could possibly be due to clinicians tending to cite a depressive symptom when referring an individual with psychosis to psychotherapy. Additionally, the severity of depressive symptoms, as well as having at least one recorded mention significantly increased likelihood of having at least one CBTp session. In the sample of those with the highest number of depressive symptoms (top 25%), relationships between CBTp receipt and comorbid anxiety diagnosis, age, gender, ethnicity, prior CBT, negative and disorganised psychotic symptoms remained (effect size ranging from 0.08 to 1.34). This suggests the importance of these predictors in a reasonable sample of patients with higher clinical need for CBTp receipt.

Overall, there was therefore a low prevalence of CBTp receipt within those with one depressive symptom. The depressive symptom which was the strongest predictor of this intervention in fully adjusted models was disturbed sleep. There is a known high prevalence of sleeping problems in this population,[20 21] described by some researchers as an 'intrinsic feature of schizophrenia,[22] known to reduce quality of life, decreasing coping and exacerbate positive symptoms.[23] The significant association between insomnia and psychotic-like symptoms, such as paranoia, has also been seen in non-clinical populations.[24] Furthering this, the recommended first line of treatment for sleep problems in this sample is CBT.[25] Poor concentration was the next strongest depressive symptom predictor in the fully-adjusted model, supporting previous research of its association with psychosis vulnerability.[26] The significance of helplessness, guilt and hopelessness mirrors CBTp research that found significant post-treatment reduction in hopelessness, self-depreciation and guilt using the Calgary Depression Rating Scale for Schizophrenia.[27] Other significant depressive symptoms associated with low self-esteem and negative self-evaluation and emotions have been found to significantly affect the development and severity of positive symptoms.[28] This may be because positive symptoms develop as a psychological defence against low self-esteem[29] and depression-induced guilt.[30] Therefore, it could be suggested that the significance of each of the depressive symptoms is often linked to psychotic symptoms and CBTp effectiveness. However, while there is evidence on the clinical impact of depressive symptoms in schizophrenia, the associations with choice of therapy must be viewed as exploratory and in need of independent replication. While a

**Table 6** Unadjusted, partially and fully adjusted logistic regression models for CBTp receipt (regression models 1, 2 and 3) with continuous symptom measures

| | N (%) | Unadjusted OR (95% CI) | Partially adjusted OR (95% CI) | Fully adjusted OR (95% CI) |
|---|---|---|---|---|
| Depressive symptoms | | | | |
| Severity | 18 286 (91.1) | 0.29 (1.31 to 1.35)*** | 0.27 (1.20 to 1.44)*** | 0.23 (0.13 to 0.33)*** |
| Positive symptoms | | | | |
| Severity | 20 078(100) | | −0.18 (0.75 to 0.92)*** | −0.21 (−0.31 to -0.09)*** |
| Bipolar diagnosis | | | | |
| Has f31 diagnosis | 959 (4.80) | | 0.21 (0.94 to 1.63) | 0.15 (−0.13 to 0.43) |
| Depression diagnosis | | | | |
| Has f32 diagnosis | 603(80) | | −0.09 (0.72 to 1.15) | −0.09 (−0.33 to 0.15) |
| Anxiety diagnosis | | | | |
| Has f40/41 diagnosis | 556 (2.80) | | 0.49 (1.15 to 2.29)*** | 0.46 (0.11 to 0.80)*** |
| Age | N/A | | −0.02 (0.97 to 0.99)*** | −0.02 (−0.03 to -0.01)*** |
| Gender | | | | |
| Male | | | Reference category | |
| Female | 8353 (41.60) | | 0.17 (0.97 to 1.44)* | 0.17 (−0.03 to 0.36) |
| Ethnic group | | | | |
| White British | 6056 (30.10) | | Reference category | |
| White other | 2048 (10.20) | | −0.41 (0.45 to 0.96)** | −0.44 (−0.93 to -0.07)*** |
| Black | 7406 (36.90) | | −0.25 (0.63 to 0.97)** | −0.29 (−0.51 to -0.07)** |
| Asian | 1279 (6.40) | | −0.66 (0.32 to 0.80)*** | −0.67 (−1.13 to -0.23)*** |
| Other/mixed | 1981 (9.90) | | −0.14 (0.62 to 1.22) | −0.16 (−0.50 to 0.18) |
| Not stated | 1308 (6.50) | | −0.92 (0.02 to 2.42) | −0.79 (−3.74 to 1.01) |
| Primary diagnosis | | | | |
| Schizophrenia | 9845 (49.00) | | Reference category | |
| Schizoaffective disorder | 2142 (10.70) | | −0.08 (0.69 to 1.21) | −0.11 (−0.40 to 0.18) |
| Other schizophrenia spectrum | 8091 (40.30) | | 0.02 (0.82 to 1.26) | −0.06 (−0.15 to 0.28) |
| Negative symptoms | | | | |
| Severity | 13 169 (65.60) | | | 0.06 (−0.01 to 0.123)* |
| Manic symptoms | | | | |
| Severity | 17 945 (89.40) | | | −0.01 (−0.13 to 0.12) |
| Disorganisation symptoms | | | | |
| Severity | 11 513 (57.30) | | | 0.10 (−0.05 to 0.25) |
| CBT prior | | | | |
| 1+ prior CBT session | 1647 (8.20) | | | 0.62 (0.34 to 0.89)*** |

Unadjusted (model 1): depressive symptom severity as a predictor with no adjusted covariates.
Partially adjusted (model 2): results were adjusted for age, ethnicity, gender, diagnostic group,f31 diagnosis, f32 diagnosis, f40/41 diagnosis, positive symptom severity.
Fully adjusted (model 3): results were adjusted for age, ethnicity, gender, diagnostic group, f31 diagnosis, f32 diagnosis, f40/41 diagnosis, positive symptom severity, prior CBT, negative symptom severity, disorganisation symptom severity and manic symptom severity.
*P<0.05, **p<0.01, ***p<0.001.
CBT, cognitive–behavioural therapy; CBTp, CBT for psychosis; NA, not available.

possibility may be that clinicians are assuming that certain depressive symptoms are likely to be more responsive to CBTp than others, there may be other unknown reasons for therapy choice that requires further investigation. General results suggest that receipt of this intervention requires an increase for all of this population before individuals with these specific symptoms could be targeted.

Regarding negative symptoms, the non-significant associations between specific negative symptoms (that overlapped with depressive symptoms) and CBTp receipt

**Table 7** Partially and fully adjusted logistic regression models for CBTp receipt with individual depressive symptoms as covariates within top 25% quantity of depressive symptoms (regression models 9 and 10)

| | N (%) | Partially adjusted OR (95% CI) | Fully adjusted OR (95% CI) |
|---|---|---|---|
| Bipolar diagnosis | | | |
| Has f31 diagnosis | 541 (10.78) | 0.19 (−0.03 to 0.41)* | 0.15 (−0.08 to 0.38) |
| Depression diagnosis | | | |
| Has f32 diagnosis | 885 (17.63) | 0.11 (−0.08 to 0.30) | 0.08 (−0.90 to 1.30) |
| Anxiety diagnosis | | | |
| Has f40/41 diagnosis | 270 (5.38) | 0.53 (0.25 to 0.80)*** | 0.47 (0.19 to 0.75)*** |
| Age | M=36.24 (18-93) | −0.02 (−0.02 to −0.01)*** | −0.02 (−0.03 to −0.01)*** |
| Gender | | | |
| Male | 2059 (41.01) | Reference category | |
| Female | 2960 (58.99) | 0.20 (0.05 to 0.34)*** | 0.17 (0.02 to 0.32)** |
| Ethnic group | | | |
| White British | 1486 (29.59) | Reference category | |
| White other | 433 (8.63) | −0.21 (−0.48 to 0.05) | −0.22 (0.50 to 0.05) |
| Black | 2262 (45.08) | −0.32 (−0.49 to −0.16)*** | −0.31 (0.47 to −0.14)*** |
| Asian | 328 (6.53) | −0.53 (0.86 to −0.21)*** | −0.52 (0.85 to −0.20)*** |
| Other/mixed | 467 (9.31) | −0.08 (−0.34 to 0.17) | −0.08 (0.34 to 0.17) |
| Not stated | 43 (0.86) | −1.42 (−2.85 to 0.40)** | −1.34 (−2.8 to −0.31)** |
| Primary diagnosis | | | |
| Schizophrenia | 2219 (44.22) | Reference category | |
| Schizoaffective disorder | 740 (14.72) | 0.04 (−0.18 to 0.26) | 0.003 (−0.22 to 0.22) |
| Other schizophrenia spectrum | 2060 (41.03) | 0.02 (−0.14 to 0.17) | 0.05 (−0.11 to 0.21) |
| Negative symptoms | | | |
| 1+ negative symptom mention | 4956 (98.7) | | −0.88 (1.41 to −0.33)*** |
| Disorganisation symptoms | | | |
| 1+ disorganisation symptom mention | 4199 (83.66) | | 1.18 (−0.02 to 0.38)* |
| CBT prior | | | |
| 1+ prior CBT session | 436 (8.7) | | −0.90 (0.68 to 1.11)*** |

Partially adjusted (model): results were adjusted for age, ethnicity, gender, diagnostic group, f31 diagnosis, f32 diagnosis, f40/41 diagnosis.
Fully adjusted (model): results were adjusted for age, ethnicity, gender, diagnostic group, f31 diagnosis, f32 diagnosis, f40/41 diagnosis, prior CBT, negative symptoms, disorganisation symptoms.
*P<0.05, **p<0.01, ***p<0.001.
CBT, cognitive–behavioural therapy; CBTp, CBT for psychosis.

requires specific further testing. This was not conducted in the current study due to the primary aim focusing on depressive symptoms. However, from our results on specific depressive symptoms in table 5, symptoms that overlapped with negative symptoms (anhedonia, anergia, apathy and blunted affect) were not associated with CBTp receipt. Additionally, negative symptoms significantly decreased likelihood of CBTp receipt within the top 25% of individuals with a depressive symptom mention. Overall, this raises concerns that individuals with these specific negative/depressive symptoms are no more likely and perhaps less likely to receive CBTp than someone without these symptoms. Possibly, this is due to clinicians not tending to refer these individuals because they do nt

believe intervention would be effective. This is in line with a CBTp review of randomised control trials, finding non-significant reductions of negative symptoms,[31] perhaps due to the narrowing of treatments to specifically target positive symptoms.[32] However, further work should be undertaken to verify that individuals are not being denied a potentially beneficial intervention because of their symptom profile.

Prior CBT receipt, comorbid disorder presence and specific symptoms (manic, disorganised and negative) also emerged as independent predictors of CBTp receipt for the general sample and within those with the top 25% depressive symptom numbers. Within table 4, individuals who had any recorded CBT receipt prior to the

index date were 1.29 times more likely to have recorded CBTp receipt later on. Also, patients with an additional comorbid disorder were 0.32–0.73 times more likely to have received CBTp compared with those with just a psychosis diagnosis. However, in the top 25% of individuals within table 7, individuals with prior CBT were 0.40 times less likely to receive CBTp. This finding requires further research to understand the effects of prior CBT and negative symptoms on CBTp receipt within different psychosis subsamples. Additionally, those with anxiety were 0.47 times more likely to receive CBTp and those with disorganised symptoms were 1.18 times more likely, respectively. After general CBTp receipt has increased, there could be a method to focus more on patients with different types of psychotic symptoms and comorbid affective diagnosis. Furthering this, future research could investigate whether those who have had prior general CBT would benefit from CBTp, or whether those who have not had any experience developing cognitive behaviour skills in therapy should be targeted.[33]

Crucially, there were also significant differences in CBTp receipt between different ethnic and gender groups. Male patients were 0.20 times less likely in the general sample and top 25% of depressive symptoms to have recorded CBTp receipt. Black, Asian, other and mixed ethnic groups were between 0.21 and 0.49 times less likely to have a documented CBTp session compared with individuals of white ethnicity within both the general and top 25% depressive symptoms samples. Inequitable access to CBTp has been identified in previous CBTp research within a psychosis sample drawn from the same data resource in 2017, finding female patients to be more likely to have received CBTp and individuals of White ethnicity to have a significantly higher likelihood of CBT receipt than black or other ethnicity groups.[11] This also supports results from a recent CBTp study focusing specifically on ethnic group differences in CBTp provision within SLaM, who found that in comparison to white British individuals, those from black ethnic groups with psychosis or bipolar disorder were significantly less likely to have a documented CBTp session. This is especially important when considering the high prevalence of psychosis within UK BAME populations.[33] Inequality in CBTp receipt may be due to ethnic variations in CBTp engagement. Some of these barriers within certain communities may be increased stigma, fear of clinicians by service-users or service users by clinicians, institutional racism within mental health services or non-culturally appropriate therapy.[34] As differences in documented CBTp receipt between ethnic groups have now been documented by three different papers in this service, it is imperative that further work is conducted to increase provision of CBTp within groups less likely to receive treatment. This may include targeted outreach programmes and culturally adapting interventions[34] within these minority groups.

This study has a number of strengths and limitations. Generally, focusing on patients with more diverse functioning, comorbidity and symptom severity levels helps research identify a larger number of predictors of clinical outcomes. This can be seen through our results, where negative, manic and disorganisation symptoms significantly predicted CBTp receipt, as well as recent heterogeneous research,[32] that was the first to identify depression as a significant predictor of positive symptom improvement post-CBTp. This highlights the importance of focusing on a clinically heterogeneous sample to realistically determine significant predictors of CBTp receipt. Second, using an NLP approach automates the measurement of what would otherwise require manually conducted audits on records and case notes, increasing the number of cases that can be investigated and providing a method that could be used more routinely to monitor CBT receipt. The large sample size enabled us to identify clinical differences in the real-life administration of CBTp within a psychosis cohort, and we were able to adjust for multiple clinical variables and comorbidity diagnoses to provide a more realistic understanding of the depressive symptom-CBTp receipt relationship. This time frame was broad to allow the inclusion of as many active patients receiving CBTp as possible, additionally circumventing monthly/seasonal variation of CBTp receipt.

One limitation of the study was the omission of strict time periods for the mention of clinical symptoms prior to CBTp administration. Unfortunately, using this approach would have involved implementing time periods on all of the other clinical symptoms and variables, which would have been difficult considering the number of variables that would need to be controlled. In addition, the NLP symptom algorithms do not currently distinguish between past or present symptoms. Therefore, symptom mentions documented after the CBTp receipt date could refer to mentions of symptoms occurring prior to CBTp receipt, reducing the effectiveness of using time periods. A follow-up time period after the index date was also not established, meaning that participants included in the cohort at a later date may have been less likely to have had a CBTp session, due to their limited time period within the service. Additionally, we did not have data regarding which type of service was providing CBTp for each patient (eg, early intervention services compared with other community services). Future studies should examine whether CBTp receipt differs depending on the service, especially considering how effective CBTp provision may be in those at ultrahigh risk.

While our use of additional querying of text fields allowed us to identify a significantly larger number of CBTp episodes than using structured data alone, we were not able to quantify the gap between CBTp referral and CBTp receipt. This is because the CBTp NLP algorithm detects CBTp receipt rather than CBTp being offered, due to the wide range of subtle wording used for the latter more complex entity. The results combine effects on the likelihood of CBTp being offered, with those on session receipt following an offer. While this may have affected our results, previous service audits have suggested that the severity and occurrence of depressive symptoms

significantly decreases CBT receipt.[35] Therefore, if only receipt was directly measured, we would expect to see similar results. Additionally, completion rates and effectiveness of the CBTp was not measured, meaning we were unable to quantify the quality or focus of the sessions. Lastly, analysis was limited to patients above 18 years old, reducing the generalisability of results to those who develop a schizophrenia-spectrum disorder after this age. However, the outcome of interest was CBTp receipt within a relatively homogeneous service structure of working age services, rather than young people treated within Child and Adolescent services. Future studies should examine whether CBTp receipt differs in these services.

## Future directions

Initiatives such as the Improving Access to Psychological Therapies programme for serious mental illness, early intervention access and projects to decrease waiting times for referral have been developed to target this clinical population. However, access still falls short of recommendations and is inequitable for specific psychotic diagnoses, age and ethnicity.[11] Therefore, given the effect of CBTp on depressive symptoms,[36] perhaps its more pragmatic to focus on patients with additional depressive symptoms. Monitoring CBTp receipt over time could decipher whether these initiatives are effective at increasing general access for those with psychosis, and specific access for different sociodemographic groups and those with additional depressive symptoms (who may benefit the most).

The significant secondary clinical and sociodemographic variables require further analysis in order to fully understand the services' provision. This could involve attention given to the independent symptoms within the negative, manic and disorganisation categories in a similar manner to the specific depressive symptom regression models analysed. Further research could also explore why the presence of comorbid anxiety and bipolar disorder in this sample predicted CBTp receipt. Additionally, the results suggest a need to reflect on the steps taken since the previous service study,[34] regarding inequality in CBTp receipt among gender and ethnic groups, due to the consistent significant results seen. Regarding the use of EHR data, future work could involve developing a separate NLP algorithm to ascertain the offering of CBTp or provide another structured field for clinicians to complete for this. However, additional text fields seem an unlikely approach, as clinicians prioritise text field data for communication about CBTp sessions for themselves and their colleagues rather than to collect structure data for the sake of research. Therefore, as previously suggested,[11] it is important to accept the mixed structured-text field approach that will remain in healthcare record data and perhaps our time is best spent in improving NLP algorithms to detect the subtleties of intervention and clinical outcome data. However, the implications of our results and their consistency 3 years after the first CBTp service paper suggest the need to use this or future algorithms for service monitoring independent of these improvements.

**Acknowledgements** This paper would not have been completed without the assistance from Dr Rob Stewart. I would also like to thank Jessica Irving for her assistance in the statistical analysis and general manuscript.

**Contributors** AM planned the protocol, analysed the data and wrote the manuscript. RS was the guarantor, providing access to the data, looked over edits and revisions on the manuscript. The data from this paper was accessed through the assistance of MP and JS. The applications used to obtain data specific to CBTp were developed by CC and DC. JI assisted in statistical analysis.

**Funding** The independent research of this paper is funded by the National Institute for Health Research (NIHR) Biomedical Research Centre at the South London and Maudsley NHS Foundation Trust and King's College London. RS is part-funded by: (1) the National Institute for Health Research (NIHR) Biomedical Research Centre at the South London and Maudsley NHS Foundation Trust and King's College London; (2) a Medical Research Council (MRC) Mental Health Data Pathfinder Award to King's College London; (3) an NIHR Senior Investigator Award; (4) the National Institute for Health Research (NIHR) Applied Research Collaboration South London (NIHR ARC South London) at King's College Hospital NHS Foundation Trust. The views expressed are those of the authors and not necessarily those of the NIHR or the Department of Health and Social Care. The corresponding author (AM) has the right to grant on behalf of all authors and does grant on behalf of all authors, an exclusive licence (or non-exclusive for government employees) on a worldwide basis to the BMJ Publishing Group to permit this article (if accepted) to be published in BMJ editions and any other BMJPGL products and sublicences such use and exploit all subsidiary rights, as set out in our licence.

**Competing interests** None declared.

**Patient and public involvement** Patients and/or the public were not involved in the design, or conduct, or reporting, or dissemination plans of this research.

**Patient consent for publication** Not applicable.

**Ethics approval** The CRIS data platform has received research ethics approval as an anonymised data resource for secondary analysis (Oxford Research Ethics Committee C, reference 18/SC/0372).

**Provenance and peer review** Not commissioned; externally peer reviewed.

**Data availability statement** Data are available on reasonable request. Due to the terms of Ethics and Information Governance approvals and clinical source of the data, CRIS datasets must remain within the South London and Maudsley NHS Foundation Trust (SLaM) firewall. All data used from this study can be made accessible on request from cris.administrator@slam.nhs.uk, subject to the setting up of an appropriate research passport or SLaM honorary contract.

**ORCID iDs**
Ava Mason http://orcid.org/0000-0001-7932-3459
Jessica Irving http://orcid.org/0000-0002-2847-6508

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
