## [Reviewer comments · BMJ Open]

ARTICLE DETAILS

TITLE (PROVISIONAL)	The association between depressive symptoms and Cognitive Behavioural Therapy receipt within a psychosis sample: A cross-sectional study.
AUTHORS	Mason, Ava; Irving, Jessica; Pritchard, Megan; Sanyal, Jyoti; Colling, Craig; Chandran, David; Stewart, Robert

VERSION 1 – REVIEW

REVIEWER	Breitborde, Nicholas The Ohio State University
REVIEW RETURNED	02-Jul-2021

GENERAL COMMENTS	1) In the first paragraph of the introduction, I am unclear what the following phrase means: “relapse to mental health services.” Are the authors referring to symptomatic relapse? Inpatient hospitalization? 2) The authors list manic symptoms under the category of “psychotic symptoms.” What is the rationale for this decision? Manic symptoms typically would be considered mood symptoms as opposed to psychotic symptoms. 3) Why were the analyses limited to people >18 years old at time of referral? Would this not miss the significant number of individuals who develop a schizophrenia-spectrum disorder prior to 18? 4) What was the VIF for positive symptoms that supported not included these in the analyses? 5) It seems like an important, but unexplored, variable in this study is time. For example, did the rate of provision of CBT change among participants during the 13-year period covered in the current dataset? 6) In the discussion, the authors review the specific depressive symptoms that were associated with greater likelihood of receipt of CBTp. While they not while these specific symptoms are important to the clinical course of schizophrenia, no discussion is provided as to why clinicians may be more likely to responded to presentation of these specific symptoms with at CBTp referral. For example, what is known about clinician behavior that could account for why sleep dysfunction was the depressive symptom most likely to elicit a referral to CBTp?
---

REVIEWER	Richardson, Thomas
-----------------	--------------------

	Solent NHS Trust
REVIEW RETURNED	30-Jul-2021

GENERAL COMMENTS	This paper appears well-written, and the implications are important. There are however some considerations and potential changes to analysis which I would consider prior to publication. Abstract  • Results page 1 line 37: I feel '8.2% received CBTP' should be changed to 'only 8.2% received CBTP' as this low rate is in itself an important finding. • Conclusion, p.1 line 45s: Perhaps say about demographic differences here for example change 'overall receipt of CBTP needs to increase' to 'overall receipt of CBTP is low and more common in certain demographic groups, and needs to be increased...' • Strengths and limitations of the study, p.2 line 7: You say about being unable to 'quantify the quality or focus of the sessions', please add to this (and into the limitations section of the discussion) that completion rates and effectiveness were also not analysed. Introduction  • Page 3/4 line 10: When discussion depressive symptoms linked to psychotic please briefly discuss the link/overlap with negative symptoms. • Page 3/4 line 19, when discussing the impact of depression e.g. on lower medication adherence you could reference a similar but smaller retrospective audit which also found that depression increased the risk of drop out from CBT for psychosis. • Page 3/4 line 28: Some of the other important goals of CBT for psychosis such as reducing distress related to hallucinations and delusions could be referenced here. • https://bpspsychub.onlinelibrary.wiley.com/doi/abs/10.1111/bjc.12222 • Please make clear in the introduction if there is much evidence that CBT improves depression symptoms in the context of psychosis. • Page 3/4 line 34: When discussing low rates of offering of CBTP is there any evidence that this is higher in Early Intervention in Psychosis (EIP) services? Similarly line 38 says that CBT interventions to not need to be recorded in minimum data set EIP has strict reporting requirements I believe the offering of CBT is part of this? • Please make clear in the introduction whether or not there has been any previous research showing a link between depression symptoms and receipt of CBT in psychosis. Method  • Page 4/5 line 5, please make clear what services this covered e.g. community mental health teams, acute, EIP etc. • Please make clear what time period these records cover. • Page 6/7 line 41. Statistical analysis (and in results section). I am unsure what repeating the results in those with the top 25% of depressive symptoms adds to the analysis and results overall. This is showing that depression increases receipt of CBT overall, and this is also the case in the most depressed patients. I am unsure what this adds. Consider removing this analysis, if you wish to keep it please explain why this analysis was conducted. Results  • Page 6/7, line 55 Participants: Please describe briefly in a sentence or two in the text the results that depression, Bipolar, anxiety and
--

- prior CBT also predicted CBT receipt in the chi-square analyses from table 2.
- Page 6/7, line 57 Participants: Please add the % for differences in gender. I know that these are referenced in table 2 but I feel they would be helpful in the text as well.
 - Page 7/8, line 21 table 1: Please change 'biop diagnosis' to 'Bipolar diagnosis'.
 - Page 7/8, line 36: You say positive psychotic symptoms could not be looked at as all patients had at least one positive psychotic symptom. Could you not have looked at this by looking at the number of positive symptoms and use this as a continuous measure of symptom severity and see if this predicts CBT receipt? This in itself would be an important and novel finding. IF however you want to keep the focus on depression then please explain why a continuous rather than categorical approach to analysis impact of positive symptoms was not used.
 - Page 9/10, line 6: Please make clear the results for model 4 and 5 are for only those who had a depressive symptoms, and give the sample size of this sub-sample analysed here.
 - Page 9/10, table 4. Similar to my comments about positive symptom severity. As well as looking at whether individual depressive symptoms could predict receipt of CBT, you could also use numbers of these symptoms reported as a measure of overall depressive symptoms severity and determine if the overall severity predicts receipt of CBT. Consider running this analysis, if not then please justify why not.
 - Table 5 page 11: As previously discussed, please consider removing analysis of top 25% as I am not sure what this adds. If you wish to include this please justify why this additional analysis is necessary and what it adds to the paper.

Discussion

- Page 11 line 50 'only 8.2% of individuals received CBTp, despite 91% having at least one depressive symptoms'. Please reword this as CBTp is offered not just for depression but for other symptoms, it should be offered to those with psychosis with or without depression.
- I believe more discussion about the low proportion offered is warranted as this is itself an important and worrying finding. Please compare this to previous findings. Also please be clear about what time period this represents (e.g. database covers a few years, can't say someone has never been offered CBT).
- Line 12 page 4 when discussing CBT and sleep in psychosis it might be worth referencing this trail (in a non-clinical population): Which showed significantly reduced paranoia after CBT for insomnia <https://www.sciencedirect.com/science/article/pii/S2215036617303280>
- Page 12 line 17 'contrasting results', please make clearer what contrasting results you are referring to.
- Page 12 line 55: Might be relevant to reference here any research showing higher prevalence of psychosis in BAME populations within the UK. I.e. This group have higher prevalence of psychosis but are less likely to be offered CBT.
- Page 13 line 28: IS there any evidence from other studies about whether demographic differences in how much CBTp is offered, not just actually delivered?
- Page 13 line 40: 'Given the effect of CBTp on depressive symptoms', please give a reference here, I don't think such a reference has been provided elsewhere in the paper.

VERSION 1 – AUTHOR RESPONSE

Reviewer 1: Dr. Nicholas Breitborde, The Ohio State University

Comment 1: In the first paragraph of the introduction, I am unclear what the following phrase means: “relapse to mental health services.” Are the authors referring to symptomatic relapse? Inpatient hospitalization?

Response: Thank you for this point. This phrase within the first introduction paragraph has been modified to ‘psychotic relapse (significant increase in psychotic symptoms)’.

Introduction paragraph 1: ‘This comorbid depression increases the likelihood of having a lower quality of life, function, motivation, poorer social relationships, lower medication adherence and psychotic relapse (significant increase in psychotic symptoms).^{8,9,10}’

Comment 2: The authors list manic symptoms under the category of “psychotic symptoms.” What is the rationale for this decision? Manic symptoms typically would be considered mood symptoms as opposed to psychotic symptoms.

Response: Thank you for raising this point. So called ‘manic’ symptoms are often present in people with psychotic disorder at a time of relapse; however, we can see that the overarching label of ‘psychotic symptoms’ is misleading. Therefore, this has been amended to ‘symptom profile’ or simply ‘symptoms’ when referring to the three scales (negative, manic or disorganization) together. To reference:

Abstract: ‘Secondary: Whether age, gender, ethnicity, symptom profiles (negative, manic and disorganisation symptoms), a comorbid diagnosis of depression...’

Introduction paragraph 4: ‘... Secondary predictors of receipt were type of psychosis diagnosis (schizophrenia, schizoaffective disorder or other schizophrenia spectrum disorder), symptom profiles (negative, manic or disorganisation)’

Methods paragraph 5: ‘NLP algorithms for each specific symptom were used to identify recorded symptom profiles within participants.’

Discussion paragraph 4: ‘However, further work should be undertaken to verify that individuals are not being denied a potentially beneficial intervention because of their symptom profile.’

Discussion paragraph 5: ‘Prior CBT receipt, comorbid disorder presence and specific symptoms (manic, disorganised and negative) also emerged as independent predictors of CBTp receipt...’

Comment 3: Why were the analyses limited to people >18 years old at time of referral? Would this not miss the significant number of individuals who develop a schizophrenia-spectrum disorder prior to 18?

Response: Thank you for raising this important point. We can see the importance of identifying younger individuals; however, the outcome of interest was the receipt of a particular type of clinical intervention (CBTp), focusing on a relatively homogenous service structure. Therefore, we investigated age groups traditionally seen by working age services rather than younger

people who would be reviewed by Child and Adolescent services in standard UK mental health providers. However, as this is an important point to mention, we have added the importance of including this group within the limitations section.

Discussion paragraph 9: 'Lastly, analysis was limited to patients above 18 years old, reducing the generalisability of results to those who develop a schizophrenia-spectrum disorder after this age. However, the outcome of interest was CBTp receipt within a relatively homogenous service structure of working age services, rather than young people treated within Child and Adolescent services. Future studies should examine whether CBTp receipt differs in these services'

Comment 4: What was the VIF for positive symptoms that supported not included these in the analyses?

Response: Our apologies for this oversight. While VIF was tested to measure multicollinearity, the reason for the exclusion of positive symptoms was because factor variables need to include at least two levels, and all participants within the regression had at least one positive symptom. Additional analysis with psychosis as a continuous variable has also been included as recommended in other reviewer comments. However, relating the VIF statement, this has now been modified:

Methods paragraph 5: 'All variables were included due to their VIF values being below five. However, positive symptoms were excluded, as this factor variable only had one level, due to all participants having at least one positive symptom.'

Comment 5: It seems like an important, but unexplored, variable in this study is time. For example, did the rate of provision of CBT change among participants during the 13-year period covered in the current dataset?

Response: Thank you for raising a very important point that we had not considered. We have now added a descriptive statistics table (table 3) and figure (figure 1) within the results section, examining specifically the frequency of CBT (prior CBT and post CBTp) receipt within each of the 13 years. This table is referenced within the methods, results, and discussion section.

Methods paragraph 9: 'Descriptive statistics were also provided for yearly CBT and CBTp receipt within the data extraction period (2007-2020).'

Results paragraph 2: 'The descriptive results shown suggest that there is a low prevalence of both prior CBT and CBTp post diagnosis across the years, with receipt reducing in recent years (2019-2020) compared to earlier years (2007) of the data extraction period.'

Results paragraph 2: 'Table 3. Distribution frequencies on CBT receipt (prior to diagnosis) and CBTp receipt (post diagnosis) per year of data extraction.'

Results paragraph 2: 'Figure 1. Graph demonstrating the frequency of general CBT receipt (CBT receipt prior to diagnosis and CBTp receipt post diagnosis) per year of extraction period.'

Discussion paragraph 1: 'This finding shows a lower overall level of recorded CBTp provision compared to previous studies in 2013 (12.8%) and 2014 (14.8%)¹². This requires further examination, considering the importance of CBTp mentioned within NICE universal access recommendations.¹³ Additionally, the significant decrease of CBTp receipt in 2020 can be explained by the COVID pandemic and therefore, it is important to consider how we can improve receipt despite this.'

Comment 6: In the discussion, the authors review the specific depressive symptoms that were associated with greater likelihood of receipt of CBTp. While these specific symptoms are important to the clinical course of schizophrenia, no discussion is provided as to why clinicians may be more likely to respond to presentation of these specific symptoms with at CBTp referral. For example, what is known about clinician behavior that could account for why sleep dysfunction was the depressive symptom most likely to elicit a referral to CBTp?

Response: Thank you for another important point to consider. We have checked the literature and have concluded that while there is research examining the effectiveness of CBTp on depressive symptoms, there are a lack of studies examining clinician behaviour on CBTp referral or receipt for those with or without depressive symptoms. Therefore, the following point has been modified:

Discussion paragraph 3: 'Therefore, it could be suggested that the significance of each of the depressive symptoms is often linked to psychotic symptoms and CBTp effectiveness'

Also, this statement has been included within the discussion:

Discussion paragraph 3: 'However, while there is evidence on the clinical impact of depressive symptoms in schizophrenia, the associations with choice of therapy must be viewed as exploratory and in need of independent replication. While a possibility may be that clinicians are assuming that certain depressive symptoms are likely to be more responsive to CBTp than others, there may be other unknown reasons for therapy choice that requires further investigation.'

Reviewer 2: Dr. Thomas Richardson, Solent NHS Trust

This paper appears well-written, and the implications are important. There are however some considerations and potential changes to analysis which I would consider prior to publication.

Abstract

- Results page 1 line 37: I feel '8.2% received CBTp' should be changed to 'only 8.2% received CBTp' as this low rate is in itself an important finding.
- Conclusion, p.1 line 45s: Perhaps say about demographic differences here for example change 'overall receipt of CBTp needs to increase' to 'overall receipt of CBTp is low and more common in certain demographic groups, and needs to be increased...'
- Strengths and limitations of the study, p.2 line 7: You say about being unable to 'quantify the quality or focus of the sessions', please add to this (and into the limitations section of the discussion) that completion rates and effectiveness were also not analysed.

Response: Many thanks for these points. We have reworded the results and conclusion sentences within the abstract. On the issue of the relatively low proportion, we feel that it's important to make clear that the observation is based on 'recorded CBTp' in as far as we were able to ascertain this. We have made numerous re-wordings of the outcome, so that this is made clearer and have added text to the abstract, limitations and discussion section, considering the potential for under-ascertainment of this intervention where it was not recorded or not recorded with clear enough wording in the EHR.

Abstract, strengths and limitations: 'Furthering this, it cannot be used to examine CBTp completion rates and effectiveness'.

Discussion paragraph 9: 'It is also important to consider that we are only ascertaining recorded CBTp receipt, which may result in failing to pick up all CBTp receipt instances. Considering that previous research describing the app development suggests high precision and recall

performance of CBTp instances (PPV= 96%, sensitivity= 96%),¹² it could be suggested that low prevalence within the results is due to lack of recording within the clinical health records, rather than lack of app identification. Therefore, stricter regulations are required for CBTp to be reported within clinical health records. Additionally, completion rates and effectiveness of the CBTp was not measured, meaning we were unable to quantify the quality or focus of the sessions. Lastly, analysis was limited to patients above 18 years old, reducing the generalisability of results to those who develop a schizophrenia-spectrum disorder after this age. However, the outcome of interest was CBTp receipt within a relatively homogenous service structure of working age services, rather than young people treated within Child and Adolescent services. Future studies should examine whether CBTp receipt differs in these services.'

Introduction

- Page 3/4 line 10: When discussing depressive symptoms linked to psychotic please briefly discuss the link/overlap with negative symptoms.^[1]^[2]^[SEP]

Response: Thank you for raising this relevant point, we have added a sentence on this within the introduction.

Introduction paragraph 1: 'Additionally, focusing on mood symptoms such as self-esteem and pessimism can help differentiate depressive symptoms from negative psychotic symptoms, that often show significant clinical overlap⁵'

- Page 3/4 line 19, when discussing the impact of depression e.g. on lower medication adherence you could reference a similar but smaller retrospective audit which also found that depression increased the risk of drop out from CBT for psychosis.^[1]^[2]^[SEP]

Response: Many thanks for providing this reference, we have added the reference within that specific introduction point:

Introduction paragraph 1: 'This comorbid depression increases the likelihood of having a lower quality of life, function, motivation, poorer social relationships, lower medication adherence and psychotic relapse ^{8,9,10}'.

- Page 3/4 line 28: Some of the other important goals of CBT for psychosis such as reducing distress related to hallucinations and delusions could be referenced here. <https://bpspsychub.onlinelibrary.wiley.com/doi/abs/10.1111/bjc.12222>^[1]^[2]^[SEP]

Response: We understand from this how important it is to specify distress reduction specifically related to hallucinations and delusions. Therefore, we have modified the introduction point and included this reference within:

Introduction paragraph 2: 'In the UK, the National Institute of Clinical Excellence¹³ has recommended that cognitive behavioural therapy for psychosis (CBTp) be offered universally to individuals with psychosis. Based on the stress-vulnerability model, ¹⁴ CBTp focuses on distress reduction related to hallucinations and delusions, through targeting negative beliefs and improving self-esteem.^{10, 15}'

- Please make clear in the introduction if there is much evidence that CBT improves depression symptoms in the context of psychosis.^[1]^[2]^[SEP]

Response: Many thanks for this comment, we have modified a key sentence within the introduction to validate this point.

Introduction paragraph 2: 'While studies examining characteristics of CBTp show strong evidence that CBTp improves depressive symptoms in the context of psychosis, specifically with long term reductions in suicidal behaviour,^{10,15,16} service provision of this intervention still falls far short of the universal access recommended.¹²'

- Page 3/4 line 34: When discussing low rates of offering of CBTp is there any evidence that this is higher in Early Intervention in Psychosis (EIP) services? Similarly line 38 says that CBT interventions do not need to be recorded in minimum data set EIP has strict reporting requirements I believe the offering of CBT is part of this?

Response: Many thanks for suggesting the important point of services providing different rates of CBTp receipt. Regarding line 38, this point was raised in the context of CBTp provision within SLAM EIP and promoting recovery community services for people with psychosis. The study (Colling et al. 2017) states that while there is a drop down text box in order to report this information within both services, it is not mandatory. Additionally, we must apologise as within our own study, we currently do not have access to that granularity of data, to examine whether service type specifically affected CBTp receipt. However, considering the importance of this point, we have mentioned this issue within the limitations.

Discussion paragraph 8: 'Additionally, we did not have data regarding which type of service was providing CBTp for each patient (for example, early intervention services compared to other community services). Future studies should examine whether CBTp receipt differs depending on the service, especially considering how effective CBTp provision may be in those at ultra-high risk.'

- Please make clear in the introduction whether or not there has been any previous research showing a link between depression symptoms and receipt of CBT in psychosis.

Response: Many thanks for highlighting this point. While this has been mentioned as a strength within the abstract ('this is the first electronic health record (EHR) study to measure how clinical symptomatology predicts CBTp receipt'), this has been made clearer within the introduction.

Introduction paragraph 4: 'While studies have examined general CBTp receipt within patients with psychosis, no study has examined a link between depressive symptoms and CBTp receipt.¹² Therefore, we investigated whether depressive symptoms predict CBTp receipt in people with psychosis by applying these previously data extraction techniques to secondary mental health care EHRs for a large South London catchment population.'

Method

- Page 4/5 line 5, please make clear what services this covered e.g. community mental health teams, acute, EIP etc.

Response: Thank you for highlighting this point, the services included have now been made clearer within the methods section.

Methods paragraph 1: 'SLaM care covers all specialist mental health care, including early intervention services, liaison and crisis teams and community and inpatient services.'

- Please make clear what time period these records cover.^[1]

Response: Within the methods section, we have mentioned the date of first EHR use within SLaM, as well as the time periods covered within the study. We apologise if this does not seem clear, and have double checked these sentences to make sure they are clear to the reader.

Methods paragraph 1: 'EHRs have been used for all SLaM services since 2006, with the Clinical Record Interactive Search system (CRIS) being established in 2008 to facilitate the retrieval of de-identified data from these records of patients previously or currently receiving mental healthcare from SLaM.'

Methods paragraph 2: 'We extracted data for all individuals receiving SLaM care between January 2007 and June 2020 with a primary diagnosis of an ICD-10-defined schizophrenia spectrum disorder (F20-F29) and above the age of 18 at the time their original referral was accepted.'

- Page 6/7 line 41. Statistical analysis (and in results section). I am unsure what repeating the results in those with the top 25% of depressive symptoms adds to the analysis and results overall. This is showing that depression increases receipt of CBT overall, and this is also the case in the most depressed patients. I am unsure what this adds. Consider removing this analysis, if you wish to keep it please explain why this analysis was conducted.

Response: Our apologies if the rationale for this additional analysis is unclear. This was conducted so that the study did not simply show that CBTp was more often received in this specific group, but to investigate the level and predictors of receipt where a clear clinical indication (in terms of depressive symptoms) was present, supplementing the findings for the cohort overall. We would prefer to retain the analyses, as this was planned a priori, however we are happy to accept an editorial judgement on the matter. Currently, additional sentences have been added within the manuscript to explain this rationale more clearly.

Methods paragraph 13: 'This subsample analysis was conducted to examine predictors of CBTp receipt where a clear clinical indication was present, supplementing the overall findings.'

Discussion paragraph 1: 'This suggests the importance of these predictors in a reasonable sample of patients with higher clinical need for CBTp receipt.'

Results paragraph 7: 'This sample comprised individuals with the top 25% number of depressive symptoms (5018 patients), defined to reflect those who might reasonably expect to receive CBT.'

. Page 6/7, line 55 Participants: Please describe briefly in a sentence or two in the text the results that depression, Bipolar, anxiety and prior CBT also predicted CBT receipt in the chi-square analyses from table 2.

. Page 6/7, line 57 Participants: Please add the % for differences in gender. I know that these are referenced in table 2 but I feel they would be helpful in the text as well.

Response: Thank you for raising these important details, they have been included within the participant paragraph.

Results paragraph 1: 'All mentioned variables showed significant between-group differences at $p < .001$ apart from gender (No CBTp delivery group females=41.4%, CBTp delivery group females=43.5%; $X^2=2.75$, $p=.097$). These significant variables include depression diagnosis ($X^2=87.36$), bipolar diagnosis ($X^2=71.94$), anxiety diagnosis ($X^2=118.28$) and prior CBT receipt ($X^2=497$).'

. Page 7/8, line 21 table 1: Please change 'biop diagnosis' to 'Bipolar diagnosis'.

Response: Our apologies for this oversight, this has been modified within the table, and 'no diagnosis' has been modified on the other disorders to remain consistent.

. Page 7/8, line 36: You say positive psychotic symptoms could not be looked at as all patients had at least one positive psychotic symptom. Could you not have looked at this by looking at the number of positive symptoms and use this as a continuous measure of symptom severity and see if this predicts CBT receipt? This in itself would be an important and novel finding. IF however you want to keep the focus on depression then please explain why a continuous rather than categorical approach to analysis impact of positive symptoms was not used.

Response: Many thanks for raising a relevant analysis point. We have included additional logistic regressions, with symptoms measured as continuous variables (depressive, manic, disorganized, positive, and negative) within unadjusted, partially adjusted and fully adjusted logistic regressions. These continuous variables present the frequency of symptoms mentioned within each symptom construct. The methods, results and discussion have been updated to include this new analysis. We have kept our current regression to stay true to our original 'a priori' design; however, we have also re-run all analysis in response to the comments and have presented new tables for the former regression models, with modified statistics within the discussion. Below are the additional comments made from the additional analysis you have suggested.

Abstract Outcome measures: 'Secondary: Whether age, gender, ethnicity, symptom profiles (positive, negative, manic and disorganisation symptoms)'

Abstract results: 'Results: Of patients with a psychotic disorder, only 8.2% received CBTp. Individuals with at least one depressive symptom recorded, depression symptom severity and 12 out of 15 of the individual depressive symptoms independently predicted CBTp receipt'

Methods paragraph 12: 'Additionally to measuring whether individual depressive symptoms could predict CBTp receipt, we also also measured whether overall depression severity predicted CBTp receipt. These logistic regression models involved converting depressive, disorganised, manic, positive and negative symptoms into a continuous variable, whereby severity reflected the number of different individual symptoms mentioned within each symptom construct. This allowed for positive symptoms to also be included within regression models. Model 6 was an unadjusted model, with depressive symptom severity as a predictor of CBTp receipt. Model 7 and model 8 were partially and fully adjusted models, controlling for the same variables as model 2 and 3, except categorising symptoms as the continuous rather than categorical variable.'

Results: 'Table 4 Unadjusted, partially and fully adjusted logistic regression models for CBTp receipt (Regression model 1,2 and 3) with categorical symptom measures.'

Results: 'Table 6 Unadjusted, partially and fully adjusted logistic regression models for CBTp receipt (Regression model 1,2 and 3) with continuous symptom measures.'

Results paragraph 6: 'General depressive symptom severity regression analysis. Results from the unadjusted (model 6), partially adjusted (model 7) and fully adjusted regression (model 8) are displayed in Table 6. Regression model 6 found that depression symptom severity significantly predicted CBTp receipt. Regarding model 7 and 8, depression symptom severity, positive symptom severity, anxiety diagnosis, and being of older age independently positive predicted CBTp receipt. Within model 8, negative symptom severity and prior CBT significantly predicted CBTp receipt.'

Discussion paragraph 1: 'In general, only 8.2% of individuals received CBTP within the 13-year timeframe of the study, showing the low prevalence of receipt despite current clinical guidelines. This finding shows a reduction in CBTP provision compared to previous studies in 2013 (12.8%) and 2014 (14.8%)¹², which was further supported by the descriptive frequency results, showing a drop in both CBT and CBTP receipt in recent years. This requires further examination, as it is unclear why receipt is decreasing considering the importance of CBTP mentioned within NICE universal access recommendations.¹³'

Discussion paragraph 2: 'Additionally, the severity of depressive symptoms, as well as having at least one recorded mention significantly increased likelihood of having at least one CBTP session.'

. Page 9/10, line 6: Please make clear the results for model 4 and 5 are for only those who had a depressive symptoms, and give the sample size of this sub-sample analysed here.^[SEP]

Response: We apologise for our lack of clarity, but Models 4 and 5 were not in the subsample of top 25% quantity of depressive symptoms. To make this clearer, we have included the sample size and directly made this point within the methods and results.

Methods paragraph 11: 'As the primary aim of the study was to investigate depressive symptoms as a predictor of CBTP receipt, we also split the depressive symptoms category into the 15 specific depressive symptoms applications within the whole sample.'

Results paragraph 4: 'Results from the unadjusted (model 4) and fully adjusted (model 5) regression analyses for each of the 15 individual depressive symptoms are displayed in Table 4 (N=20078). Each symptom refers to presence of at least one mention in the patients notes compared to no mention.'

Results table 1: 'Unadjusted and fully adjusted logistic regression models for CBTP receipt with individual depressive symptoms as covariates (Regression model 4 and 5) for the overall sample.'

- Page 9/10, table 4. Similar to my comments about positive symptom severity. As well as looking at whether individual depressive symptoms could predict receipt of CBT, you could also use numbers of these symptoms reported as a measure of overall depressive symptoms severity and determine if the overall severity predicts receipt of CBT. Consider running this analysis, if not then please justify why not

Response: Many thanks for highlighting this point concerning symptom severity of both depression and positive symptoms. We have added analyses as described in detail above.

- Table 5 page 11: As previously discussed, please consider removing analysis of top 25% as I am not sure what this adds. If you wish to include this please justify why this additional analysis is necessary and what it adds to the paper.^[SEP]

Response: We hope that our previous comment on this point has made our rationale clearer, and we are happy to accept an editorial judgement on the matter. Additional sentences have been included within the manuscript as detailed in response to the former comment.

Discussion^[SEP]

• Page 11 line 50 'only 8.2% of individuals received CBTp, despite 91% having at least one depressive symptoms'. Please reword this as CBTp is offered not just for depression but for other symptoms, it should be offered to those with psychosis with or without depression.^[1]

Response: This is a very helpful point; thank you for bringing it to our attention. We have modified this sentence within the discussion.

Discussion paragraph 1: 'In general, only 8.2% of individuals received CBTp within the 13-year timeframe of the study, showing the low prevalence of receipt despite current clinical guidelines.'

• I believe more discussion about the low proportion offered is warranted as this is itself an important and worrying finding. Please compare this to previous findings. Also please be clear about what time period this represents (e.g. database covers a few years, can't say someone has never been offered CBT).

Response: We have modified the discussion mention of CBTp receipt to reflect the timeframe, which we agree should not be generalised to the lifetime of the patient. Additionally, we have made more references to current clinical guidelines and the importance of research to further understanding this finding.

Discussion paragraph 1: 'In general, only 8.2% of individuals received CBTp within the 13 year timeframe of the study, showing the low prevalence of receipt despite current clinical guidelines. This finding shows a reduction in CBTp provision compared to previous studies in 2013 (12.8%) and 2014 (14.8%)¹², which was further supported by the descriptive frequency results, showing a drop in both CBT and CBTp receipt in recent years. This requires further examination, as it is unclear why receipt is decreasing considering the importance of CBTp mentioned within NICE universal access recommendations.¹³'

^[1]• Line 12 page 4 when discussing CBT and sleep in psychosis it might be worth referencing this trial (in a non-clinical population): Which showed significantly reduced paranoia after CBT for insomnia <https://www.sciencedirect.com/science/article/pii/S2215036617303280>^[1]

Response: Thank you for highlighting this important study. We have included it as a reference within the discussion.

Discussion paragraph 3: 'There is a known high prevalence of sleeping problems in this population,^{21,22} described by some researchers as an 'intrinsic feature of schizophrenia,²³ known to reduce quality of life, decreasing coping and exacerbate positive symptoms.²⁴ The significant association between insomnia and psychotic-like symptoms, such as paranoia, has also been in non-clinical populations.²⁵ Furthering this, the recommended first line of treatment for sleep problems in this sample is CBT.²⁶

• Page 12 line 17 'contrasting results', please make clearer what contrasting results you are referring to.

Response: Many thanks for raising this point; the sentence has been modified within the discussion to make this clearer.

Discussion paragraph 4: 'Regarding negative symptoms, the non-significant associations between specific negative symptoms (that overlapped with depressive symptoms) and CBTp receipt requires specific further testing. This was not conducted in the current study due to the primary aim focusing on depressive symptoms.'

- Page 12 line 55: Might be relevant to reference here any research showing higher prevalence of psychosis in BAME populations within the UK. I.e. This group have higher prevalence of psychosis but are less likely to be offered CBT. [SEP]

Response: This point expresses the importance of increased CBTp receipt within the BAME population and has been added to the discussion.

Discussion paragraph 6: 'This also supports results from a recent CBTp study focusing specifically on ethnic group differences in CBTp provision within SLAM, who found that in comparison to White British individuals, those from Black ethnic groups with psychosis or bipolar disorder were significantly less likely to have a documented CBTp session. This is especially important when considering the high prevalence of psychosis within UK BAME populations. ³⁵'

- Page 13 line 28: Is there any evidence from other studies about whether demographic differences in how much CBTp is offered, not just actually delivered? [SEP]

Response: Unfortunately, only one other study has reported findings on the association between ethnicity and CBTp. However, this study also used electronic health records from SLAM, and therefore only focused on CBT delivery, not it being offered. This could potentially be due to the non-mandatory nature of CBT receipt, let alone suggestion of CBT within the records. We have mentioned the limitation of only measuring and having data on CBTP receipt in our study within the discussion.

Discussion paragraph 9: 'While our use of additional querying of text fields allowed us to identify a significantly larger number of CBTp episodes than using structured data alone, we were not able to quantify the gap between CBTp referral and CBTp receipt. This is because the CBTp NLP algorithm detects CBTp receipt rather than CBTp being offered, due to the wide range of subtle wording used for the latter more complex entity. The results combine effects on the likelihood of CBTp being offered, with those on session receipt following an offer. While this may have affected our results, previous service audits have suggested that the severity and occurrence of depressive symptoms significantly decreases CBT receipt. ³⁵ Therefore, if only receipt was directly measured, we would expect to see similar results.'

- Page 13 line 40: 'Given the effect of CBTp on depressive symptoms', please give a reference here, I don't think such a reference has been provided elsewhere in the paper. [SEP]

Response: Our apologies for this oversight, a reference has now been added to this statement.

Discussion paragraph 10: 'Therefore, given the effect of CBTp on depressive symptoms³⁷, perhaps its more pragmatic to focus on patients with additional depressive symptoms'

VERSION 2 – REVIEW

REVIEWER	Richardson, Thomas Solent NHS Trust
REVIEW RETURNED	10-Mar-2022
GENERAL COMMENTS	Thank you for addressing my comments comprehensively. The only one change I would like prior to accepting is a minor change to one of the revisions you made:

	"This comorbid depression increases the likelihood of having a lower quality of life, function, motivation, poorer social relationships, lower medication adherence and psychotic relapse 8,9,10" The new paper you reference (10) is not on any of these issues, it shows specifically that depression increases the risk of drop out from CBT for psychosis, please amend. Thank you.
--	--

VERSION 2 – AUTHOR RESPONSE

Comment: Thank you very much for accepting my recent major revisions and pointing out this issue. This reference has been excluded from this point, with relevant reference numbers modified.